# An Evaluation of the Peri-Implant Tissue in Patients Starting Antiresorptive Agent Treatment After Implant Placement: A Nested Case–Control Study

**DOI:** 10.3390/medicina61081348

**Published:** 2025-07-25

**Authors:** Keisuke Seki, Ryo Koyama, Kazuki Takayama, Atsushi Kobayashi, Atsushi Kamimoto, Yoshiyuki Hagiwara

**Affiliations:** 1Implant Dentistry, Nihon University School of Dentistry Dental Hospital, Tokyo 101-8310, Japan; koyama.ryo@nihon-u.ac.jp (R.K.); deka25021@g.nihon-u.ac.jp (K.T.); kamimoto.atsushi@nihon-u.ac.jp (A.K.); hagiwara.yoshiyuki@nihon-u.ac.jp (Y.H.); 2Department of Comprehensive Dentistry and Clinical Education, Nihon University School of Dentistry, Tokyo 101-8310, Japan; deat25489@g.nihon-u.ac.jp; 3Department of Oral and Maxillofacial Surgery II, Nihon University School of Dentistry, Tokyo 101-8310, Japan; 4Department of Partial Denture Prosthodontics, Nihon University School of Dentistry, Tokyo 101-8310, Japan

**Keywords:** bisphosphonate, denosumab, osteoporosis, peri-implant MRONJ, risk factor

## Abstract

*Background and Objectives*: We wished to evaluate the effect of antiresorptive agents (ARAs) on peri-implant tissues and to examine the risk factors for peri-implant medication-related osteonecrosis of the jaw (MRONJ). *Materials and Methods*: The study cohort consisted of patients who underwent implant surgery or maintenance treatment between March 2012 and December 2024. The patients were divided into two groups: those in whom bisphosphonates (BPs) or denosumab (Dmab) was used to treat osteoporosis after implant treatment (the ARA group) and a control group. Peri-implant clinical parameters (implant probing depth (iPPD), implant bleeding on probing (iBoP), marginal bone loss (MBL), and mandibular cortical index (MCI)) measured at the baseline and at the final visit were statistically evaluated and compared in both groups. Risk factors were examined using a multivariate analysis of adjusted odds ratios (aORs). *Results*: A total of 192 implants in 61 patients (52 female, 9 male) were included in this study. The ARA group consisted of 89 implants (22 patients). A comparison of the clinical parameters showed that the ARA group had significantly higher variations in their maximum iPPD and iBoP values over time than those in the control group. Risk factors for peri-implantitis as objective variables were the use of ARAs (aOR: 3.91; 95% confidence interval [CI]: 1.29–11.9) and the change in the maximum iPPD over time (aOR: 1.86; 95% CI: 0.754–4.58). *Conclusions*: During long-term implant maintenance treatment, patients’ health and medication status change. Monitoring peri-implantitis, the presumed cause of peri-implant MRONJ, is essential, especially in patients who started ARA treatment after implant placement, and special attention should be paid to changes in implant pocket depth.

## 1. Introduction

Medication-related osteonecrosis of the jaw (MRONJ) is an intractable disease caused mainly by bone resorption inhibitors used in the treatment of osteoporosis and cancer metastasis. Bisphosphonate-induced osteonecrosis of the jaw was first reported by Marx et al. in 2003 [1]. Severe lesions may result in pathologic fractures of the jawbone, and extensive lesions may require alveolar osteotomy, which significantly reduces the patient’s quality of life. There are various reports on the incidence of MRONJ, and although it is rare, its incidence is expected to increase with an aging society [2,3,4]. Although the etiology, pathophysiology, and treatment of MRONJ remain unclear, these topics are gradually being elucidated [5,6,7]. Possible causative agents include anti-osteoporosis medications (AOMs), angiogenesis inhibitors, hormones, and methotrexate, but the current evidence is limited to bisphosphonates (BPs) and denosumab (Dmab) [5]. For the prevention of fragility fractures, antiresorptive agents (ARAs) such as BP and Dmab are highly effective. They significantly reduce the relative risk and are effective in reducing fractures in patients at risk of fragility fractures [8]. Therefore, BPs are the first-line therapy in the treatment of osteoporosis [9], and it can be assumed that the number of patients using BPs will increase among aging patients treated with implants [10]. Invasive surgical procedures, such as tooth extraction, were previously considered to be the main pathogenic factor in MRONJ, but more recently, the persistence of advanced inflammatory lesions has been considered problematic as a trigger for the development of the disease [5]. However, previous reports on MRONJ and implants have been limited, with most topics examining the safety of invasive implant placement procedures and their impact on osseointegration [11,12,13,14]. Recently, peri-implantitis as an infectious disease has been attracting attention, and notably, peri-implant MRONJ has been described as osteonecrosis of the jaw that has developed from peri-implantitis [15,16,17]. This suggests that the presence of persistent inflammation around an implant (peri-implantitis) may be a risk factor for MRONJ. Interestingly, these cases of MRONJ developed after implant placement in patients who started using ARAs during long-term stable implant function. Thus, in recent years, studies have investigated the presence of implants themselves as a risk; this is a problem that needs to be resolved as soon as possible in the field of implant treatment for older people [18,19,20,21].

Little attention has been paid to the impact of ARA use on the existing functional peri-implant tissues. Furthermore, the scientific research exploring the relationship between local inflammation around implants and the development of MRONJ is still heterogeneous and incomplete. To fill the evidence gap that exists, we, as periodontists, need to gather and discuss these problems. Implant treatment in patients with osteoporosis, which is increasing in our aging society, will become more complicated in the future, necessitating an improvement in the predictability of implant treatment. The purpose of this nested case–control study was to evaluate the impact of the use of ARAs on the peri-implant tissues of existing, functioning implants. The Patients, Intervention, Comparison, Outcome (PICO) framework for this study is as follows: P: patients treated with implants at a single facility; I: patients who started using ARAs after implant placement; C: patients not using ARAs; and O: verification of the differences in the clinical parameters of the peri-implant tissues between the two groups. The null hypothesis in this study is that “ARA treatment initiated under implant function does not affect changes in the clinical parameters of peri-implant tissues”. Furthermore, the ultimate goal of this preliminary study was to elucidate whether existing peri-implant tissue inflammation is an independent etiological factor in MRONJ.

## 2. Materials and Methods

### 2.1. The Study Design and Participants

This observational study was approved by the Ethics Committee of the Nihon University School of Dentistry (approval number EP23D027) and conducted according to the 1975 Declaration of Helsinki, as revised in 2013 [22], and the guidelines for observational and descriptive studies in epidemiology. The study cohort consisted of patients who attended the Implant Dentistry Department at Nihon University Dental Hospital between March 2012 and December 2024. Information about the purpose of the research and the methods used was posted on the dental hospital’s website and disclosed to the subjects.

### 2.2. Criteria for Case Selection

To avoid bias due to interoperator differences, this study included patients whose implant treatment was completed by a single periodontist (K.S.). After periodontitis was diagnosed and periodontal treatment was performed as necessary, implant surgery, prosthetic treatment, and maintenance were managed by the same periodontist. Inclusion criteria were as follows: (1) patients who had previously undergone implant placement in our clinic, (2) patients who were over 40 years of age at the time of implant placement, (3) patients who had not undergone AOM treatment prior to implant placement, and (5) patients who had received regular maintenance treatment for at least 1 year after superstructure placement.

Exclusion criteria were as follows: (1) no record of peri-implant clinical parameters at the time of maintenance treatment; (2) no dental radiographs or digital panoramic radiographs (DPR) at the time of re-evaluation; (3) no morphologic diagnosis of the mandibular inferior cortical bone due to obstacle shadows or poor positioning in DPR images; (4) patients who had previously undergone mandibular resection or reconstruction or had a history of bone destruction due to tumor lesions; (5) treatment of osteoporosis or solid cancer that developed after implant placement using only modifiers such as selective estrogen receptor modulators (SERMs), parathyroid hormone, or new active vitamin D3; (6) general contraindications such as pregnancy, metabolic disease, or immunosuppression; and (7) patients who had previously received radiation therapy to the head and neck (Figure 1).

### 2.3. Diagnosis of Periodontitis

All cases were diagnosed on the basis of the new classification by the American Academy of Periodontology (AAP) and the European Federation of Periodontology (EFP) [23,24]. Patients whose initial diagnosis was made before 2017 were diagnosed using the 1999 classification [25], and thus, these patients were rediagnosed using the new classification using data from that time. The stage classification (I–IV) was based on the disease severity and treatment complexity, and the grading categories (A, B, and C) were expressed in terms of the rate of periodontitis progression, risk of progression, response to periodontal therapy, and systemic status. After diagnosis, the initial periodontal treatment, periodontal surgery, and prosthetic treatment were performed as necessary, and finally, maintenance treatment was continued.

### 2.4. Clinical Examinations

The clinical parameters of the peri-implant tissues were recorded twice, once at the start of maintenance as a baseline and once at the last visit for re-evaluation. Implant probing pocket depth (iPPD) was measured using a periodontal pocket probe (11 COLORVUE^®^ PROBE KIT, Hu-Friedy, Chicago, IL, USA) at a probing pressure of 0.15 N. Each implant pocket was measured at six points in total in mm; the mean value was calculated, and the deepest point was recorded. Implant bleeding on probing (iBoP) was evaluated 10 s after probing the peri-implant pocket. Bleeding was quantified (no bleeding, score 0; bleeding present, score 1), and the average of 6 points was calculated for each implant (minimum score: 0, maximum score: 1). The marginal bone loss (MBL) around the implant was calculated using parallel dental radiographs (110 kV and 1–20 mA X-ray irradiation, resulting in an effective dose of 100 μSV) taken with an indicator. Imaging software (TechM@trix, SDS Viewer) was used to measure the distance from the mesial or distal platform of the implant to the most coronal osseointegration site. The average of these two values was defined as the amount of bone resorption per implant. The mean and maximum iPPD, iBoP, and MBL were measured at the baseline and at the final visit. For all parameters, changes in the mean and maximum values over time were calculated by subtracting the baseline values from the values at the final visit. DPR images at final imaging in all cases were classified into three types of mandibular inferior cortical bone morphology [26]. The pre-trained dentist (K.S.) evaluated the data twice, and the results of the second evaluation were used. The MCI was evaluated on the left and right sides, and the more severe value was ultimately used as the representative value as an index for the individual patient. The clinical data collection and analysis were performed independently by two operators, who were assessed for intra- and interobserver reproducibility using the Cohen’s kappa score. The statistical analysis of the final dataset was performed by a second operator who was blinded to the purpose and methodology of this study.

### 2.5. Diagnosis of Peri-Implantitis

Peri-implantitis was diagnosed based on the AAP/EFP statement, taking into account inflammation of the peri-implant mucosa and progressive resorption of the supporting bone [27]. Specifically, a diagnosis of peri-implantitis was made when the probing depth was 6 mm or greater at the follow-up appointment, when there was drainage or bleeding during probing, and when radiographic evidence showed that more than 25% of the length of the implant had experienced bone resorption. K.S. performed the peri-implantitis treatment based on the previously proposed protocol [28].

### 2.6. Data Sources

Data sources were extracted from the records of the first visit, including information about age, sex, body mass index (BMI), history of fragility fracture (excluding trauma before adolescence) [29,30,31], and smoking habits. From the records during maintenance treatment, the following information was extracted: AOM treatment initiated after implant function, implant size and placement site, superstructure (cemented or screw-fixed), MCI classification, peri-implant clinical parameters (iPPD, iBoP, MBL), peri-implantitis, and MRONJ development. Clinical parameters were initially evaluated at the time of superstructure placement; however, if this record was not available, the earliest point after the placement of the superstructure was used as the baseline, and the time of the last visit was used as the endpoint. The observation period was calculated from the date the superstructure was placed to the last visit. For implants that failed and were removed due to infection or fracture, the data recorded immediately before implant removal were used.

### 2.7. Group Selection and Matching

On the basis of the criteria of this study, the case group (the ARA group) was defined as those who started using BPs or Dmab after implant placement. Patients that used only modifying drugs such as SERMs, parathyroid hormone, vitamin D3, or combinations of these drugs for osteoporosis other than the two ARAs were excluded. Age (cases ±5 years), sex, and smoking history were estimated as potential confounding variables, and two controls (from the control group) per case were individually countermatched based on these variables.

### 2.8. Statistical Analysis

A statistical analysis was performed using EZR (Saitama Medical Center, Jichi Medical University), which is a graphical user interface for R (version 4.0.0; The R Foundation for Statistical Computing, Vienna, Austria) [32]. Demographic data from the ARA and control groups were used to test for statistical differences in each variable. A two-tailed test with an alpha error of 0.05 and a power of 0.8 revealed that the required sample size was 10 patients for the ARA group and 20 patients for the control group. After testing the normality of the data distribution for continuous variables (age, BMI, observation period, implant diameter, implant length, IPPD, mean and maximum IBoP, and IMBL) using the Kolmogorov–Smirnov test (normally distributed at *p* ≥ 0.05), a Student’s *t*-test or the Mann–Whitney U-test was performed. For the categorical variables (sex, history of fragility fracture, smoking habit, history and diagnosis of periodontitis, MCI classification, maxilla or mandible, additional surgery, peri-implantitis, implant failure, peri-implant MRONJ, and superstructure retention), a chi-square test or Fisher’s exact test was used. To examine risk factors, we first extracted explanatory variables showing the differences between the two groups. Then, we performed a univariate analysis using peri-implantitis as the objective variable and calculated the crude odds ratio (OR) and the 95% confidence interval (CI). Additionally, to measure the association between the predictor and outcome variables while controlling for confounding factors, we calculated the adjusted odds ratios (aORs) using logistic regression models. We considered *p* < 0.05 to indicate statistical significance.

## 3. Results

### 3.1. Demographic Data

The patient demographics are shown in Table 1. A total of 61 patients (22 in the ARA group and 39 in the control group) were finally included in this study. Ten patients (16.4%) had a history of fragility fracture—significantly more in the ARA group (9) than in the control group (1). Overall, eight patients (13.1%) had a smoking habit, including a history of past smoking. The mean BMI was 21.5 ± 2.6 (15.5–29.4) overall, and five patients (8.2%) had a BMI greater than 25. Thirty-five patients (57.4%) had a history of moderate or severe periodontitis, which was significantly more common in the ARA group. However, the comparison of the individual severity (stage) and progression (grade) showed no significant differences between the two groups. The MCI was class 1 in 9 patients (14.8%), class 2 in 34 patients (55.7%), and class 3 in 18 patients (29.5%). A comparison by subgroup revealed that class 1 was significantly more prevalent in the control group (23.1%), while class 3 was significantly more prevalent in the ARA group (50%). The mean duration of ARA use was 4.1 ± 3.6 years (range: 1–16 years; median: 2.5 years).

### 3.2. Implant-Based Data Evaluation

Table 2 shows the information at the implant level for each group. There were no significant differences between the two groups in superstructure type or implant dimensions. The split crest technique was significantly more common in the control group during additional surgeries. The overall mean observation period for implants was 13.2 ± 6.4 years. This period was significantly longer in the ARA group (14.7 years) than in the control group (11.8 years). Overall, the prevalence of peri-implantitis was 18.2% (35 out of 192 implants) at the implant level and 13.1% (8 out of 61 patients) at the patient level. The prevalence was significantly higher in the ARA group (28.1% within the group at the im-plant level). Equally, 13 (6.8%) of the total 192 failed implants occurred during the study period. Peri-implant MRONJ developed in two patients in the ARA group (four implants in total) (Figure 2). The details of the ARA group are shown in Table 3.

The clinical parameters of the peri-implant tissues as primary outcomes are shown in Table 4. At the baseline, the mean iPPD in the ARA group (3.1 ± 0.6 mm) was significantly larger than that in the control group. At the last visit, the mean iPPD (3.8 ± 2.0 mm), the maximum iPPD (4.6 ± 2.3 mm), and the mean iBoP (0.3 ± 0.3) were significantly larger in the ARA group. The variation (the difference between the last visit and the baseline) was significantly larger in the ARA group than that in the control group for maximum iPPD (1.1 ± 2.0 mm) and iBoP (0.2 ± 0.3 mm). The changes from the baseline to the last visit were 2.9 ± 0.5 mm to 3.4 ± 1.8 mm for the mean iPPD, 3.5 ± 1.0 mm to 4.3 ± 2.1 mm for the maximum iPPD, 0.2 ± 0.2 to 0.2 ± 0.3 for the mean iBoP, and 1.2 ± 0.9 mm to 2.1 ± 2.3 mm for the MBL. The Mann–Whitney U-test showed that there were significant increases from the baseline to the last visit in the maximum iPPD (*p* = 0.0006) and MBL (*p* = 0.0003).

### 3.3. Assessment of Risk Factors

Among the explanatory variables, continuous variables were categorized using cutoff values. The cutoff value was set by referring to the overall mean value of the ARA group and the control group combined. The cutoff values are as follows: follow-up period (≥14 years: 1, <14 years: 0), variation in maximum iPPD (≥0.8 mm: 1, <0.8 mm: 0), and variation in iBoP (≥0.1 mm: 1, <0.1 mm: 0). Only the MCI used class for the cutoff values (Class 3: 1; Class 1 and 2: 0). After performing a univariate analysis with peri-implantitis as the objective variable, significant differences were found in the use of ARAs (OR: 3.63), history of fragility fracture (OR: 0.89), history of periodontitis (OR: 8.17), MCI class 3 (OR: 1.97), additional surgery (OR: 0.57), the duration of observation (OR: 2.58), variation in the maximum IPPD (OR: 7.42), and variation in the IBoP (OR: 2.85). A multivariate analysis was performed to examine the adjusted odds ratios. The results showed significant differences in the use of ARAs (adjusted odds ratio [aOR]: 3.91) and the change in the maximum IPPD (aOR: 5.90). No significant differences were found for the other independent variables (Table 5).

## 4. Discussion

This was a single-center nested case–control study that observed the changes in the peri-implant tissue parameters in patients who started ARA treatment after their implants were functioning. Differences between the two groups in their demographic data were observed in age, history of fragility fractures, history of periodontitis, and MCI. In this study design, we attempted to countermatch to achieve a 1:2 ratio between the ARA group and the control group in terms of age, sex, and smoking habits; however, the conditions for age and smoking habits were not completely matched. For this reason, the number of cases in the control group was insufficient to reach twice the number of cases in the ARA group, and it was assumed that this was reflected in the age difference. No differences were observed in the history of periodontitis between individual stages or grades, but overall, its prevalence was significantly higher in the ARA group. Previous systematic reviews have reported that postmenopausal women with osteoporosis tend to have a high clinical attachment level [33], and it was inferred that the cohort in this study had a similar risk. Significant deterioration was observed in the ARA group when comparing the MCI class classifications. The results of this study support recent systematic reviews suggesting that the MCI is a useful screening tool for patients with a low bone mineral density [34,35].

In the implant-based analysis, no differences were observed between the groups in terms of the implant site or implant dimensions; however, the rate of additional surgery, particularly the split crest technique, was significantly higher in the control group. The reason for this may be that the control group was in better health and able to tolerate additional surgery, but the information collected in this study is limited and can only be speculated upon. Peri-implant inflammation and subsequent implant failure were significantly higher in the ARA group. These results also support our previous findings, suggesting that systemic diseases are risk factors for peri-implant inflammation [36]. However, recent systematic reviews have reported contrary results, stating that BP use does not affect implant failure rates, and further evidence is expected to be established in the future [37]. Regarding peri-implant MRONJ, the main objective of this study, the authors witnessed cases of peri-implant MRONJ in the past that may have been related to tocilizumab [38], but these cases did not meet the inclusion criteria and were not included in this study’s cohort. In our study of the clinical parameters of the implants, we focused on the amount of change over time in reference to the new classification of periodontal diseases [23,24]. The results showed that the variations in the maximum iPPD and iBoP were significantly larger in the ARA group. These results directly indicate that peri-implant tissue changes worsened in the ARA group, leading us to reject our null hypothesis that “ARA treatment initiated under implant function does not affect changes in the clinical parameters of peri-implant tissues”.

A univariate analysis revealed significant differences in all eight examined explanatory variables when peri-implantitis was the objective variable. The ORs were greater for history of periodontitis (8.17), change in iPPD (7.42), and the use of ARAs (3.63), in that order. In the adjusted multivariate analysis, the variations in the maximum iPPD (aOR = 5.90, *p* < 0.01) and use of ARAs (aOR = 3.91, *p* < 0.05) were significant. Although systematic reviews and meta-analyses of the risk factors for peri-implantitis related to systemic diseases have been published [39,40], the strength of our study is that it is the first study, to our knowledge, to examine the risk of peri-implantitis in patients using ARAs after implant placement. Although the relationship between osteoporosis and implant failure is unclear [41,42], it is recommended that the treatment for osteoporosis, including the use of BP preparations, continue during the healing period after implant placement [43].

Our findings suggest that careful monitoring of the implant pocket depth during maintenance treatment is an important factor in determining the predictability of peri-implantitis in implant patients using ARAs. The findings of this study will contribute to the development of personalized treatment plans for each patient in the future, improving the diagnosis and prognosis of implant therapy for ARA users. One of the limitations of this study is the small number of cases in which Dmab was used. To clarify the status of patients using ARAs in an observational study, the study design should either include only BPs or increase the sample size of Dmab users. These modifications will be the subject of future studies.

## 5. Conclusions

ARA users showed greater changes in their maximum implant probing depth and bleeding from peri-implant tissues, suggesting a persistent inflammatory state. Our investigation of the risk factors for peri-implantitis suggests that the use of ARAs, including BPs and Dmab, was a risk factor, in addition to the amount of variation in the implant probing depth. When patients with implants are treated using ARAs, the predictability of the treatment outcomes may be increased by paying particular attention to local inflammation, which is presumed to cause peri-implant MRONJ.

## Figures and Tables

**Figure 1 medicina-61-01348-f001:**
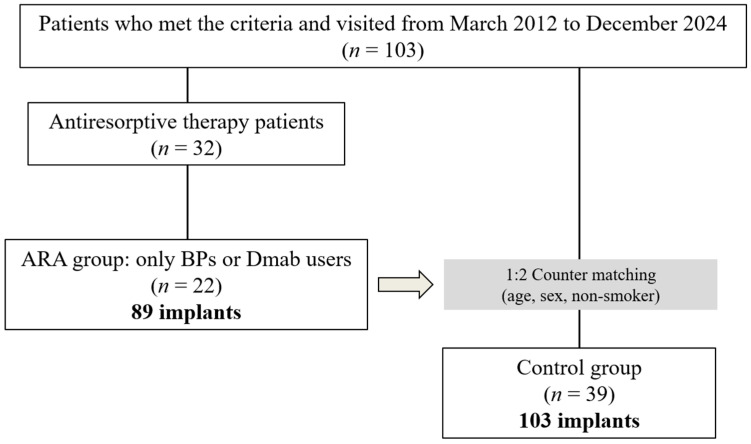
A flowchart of the study population. ARA, antiresorptive agent; BP, bisphosphonate.

**Figure 2 medicina-61-01348-f002:**
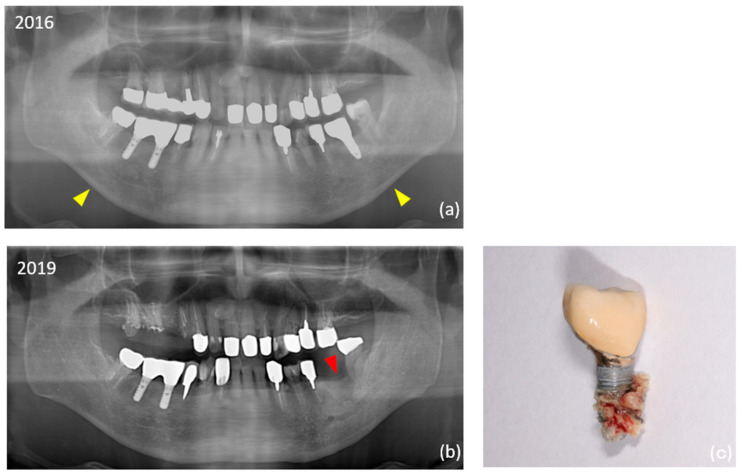
Digital panoramic radiography imaging findings in an 88-year-old man (Case 20). (**a**) The #36 implant was placed in 2010. The mandibular cortical index (MCI) classification was class 2 (yellow arrowhead). Class 1: a smooth inner surface of the cortical bone; class 2: an irregular inner surface of the cortical bone with linear resorption; class 3: severe linear resorption and cortical bone rupture over the entire cortical bone. (**b**) The patient started using an antiresorptive agent in 2017. The implant was removed in 2019 (red arrowhead). (**c**) The patient was diagnosed with stage 2 peri-implant medication-related osteonecrosis of the jaw (MRONJ).

**Table 1 medicina-61-01348-t001:** Demographic and patient characteristics for ARA group and control group.

			ARA (*n* = 22)	Control (*n* = 39)	*p* Value	Significant Difference
^a^	Sex					
		Female	19 (86.4)	33 (84.6)	1.000000	ns
^b^	Age (years)		78.4 ± 6.8	70.9 ± 8.4	0.000694	**
^b^	Body mass index (kg/m^2^)		22.0 ± 2.6	21.3 ± 2.7	0.352000	ns
^a^	History of insufficiency fracture		9 (40.9)	1 (2.6)	0.000222	**
^a^	Smoking		3 (13.6)	5 (12.8)	1.000000	ns
^a^	History of periodontitis		17 (77.3)	18 (46.2)	0.030100	*
	^c^	Stage 2 Grade A	4	3	0.240000	ns
	^c^	Stage 3 Grade A	4	5	0.710000	ns
	^c^	Stage 3 Grade B	3	6	1.000000	ns
	^c^	Stage 4 Grade B	6	3	0.059600	ns
	^c^	Stage 4 Grade C	0	1	1.000000	ns
	Mandibular cortical index					
	^c^	Class 1	0 (0)	9 (23.1)	0.020300	*
	^a^	Class 2	11 (50.0)	23 (59.0)	0.595000	ns
	^a^	Class 3	11 (50.0)	7 (17.9)	0.017700	*
	Antiresorptive therapy *overlapping					
		BPs (oral 18, injection 1)	21			
		Denosumab	3			
	Duration of antiresorptive therapy (years)		4.1 ± 3.6 (1–16)			
		1–3 years	13			
		over 4 years	11			

Mean ± S.D., or n (%), ^a^ Chi-square test, ^b^ Student’s *t*-test, ^c^ Fisher’s exact test; ARA, antiresorptive agent; BPs, bisphosphonates; ns, not significant; * *p* < 0.05; ** *p* < 0.01.

**Table 2 medicina-61-01348-t002:** Implant characteristics for ARA group and control group.

			ARA (*n* = 89)	Control (*n* = 103)	*p* Value	Significant Difference
^a^	Implant site					
		Maxilla	41 (46.1)	56 (54.4)	0.311000	ns
		Mandible	48 (53.9)	47 (45.6)		
	Implant dimensions					
	^b^	Diameter	4.0 ± 0.5	4.2 ± 0.4	0.054700	ns
		(range, median)	(3.3–5, 4)	(3.3–5, 4.1)		
	^b^	Length	10.2 ± 1.6	10.4 ± 1.5	0.294000	ns
		(range, median)	(7–16, 10)	(7–13, 10)		
^a^	Additional surgery		2 (2.3)	15 (14.6)	0.003810	*
	^a^	Sinus floor elevation	2 (2.3)	8 (7.8)	0.110000	ns
	^a^	Split crest technique	0 (0)	7 (6.8)	0.015600	*
^a^	Peri-implantitis		25 (28.1)	10 (9.7)	0.001290	**
^a^	Failure		10 (11.2)	3 (2.9)	0.045400	*
^a^	Peri-implant MRONJ		4 (4.5)	0 (0)	0.044500	*
	Superstructure					
	^a^	Cement retention	29 (32.6)	38 (36.9)	0.548000	ns
		Screw retention (including side-screw system)	60 (67.4)	65 (63.1)		
^b^	Mean maintenance duration (years)		14.7 ± 6.4	11.8 ± 6.0	0.004330	**
		(range, median)	(5–27, 13)	(2–20, 14)		

Mean ± S.D., n (%), ^a^ Chi-square test, ^b^ Mann-Whitney U test; MRONJ, medication-related osteonecrosis of the jaw; ns, not significant; * *p* < 0.05, ** *p* < 0.01.

**Table 3 medicina-61-01348-t003:** Medication data for ARA group.

Case	Age	Sex	Periodontitis		Medication	Type	Interval	Duration (Years)	Type of Implant (n)	ONJ (Location: FDI)
			Stage	Grade						
1	86	Female	3	A	Alendronate	Oral	Daily	4	Srerioss (4)	
2	85	Female	4	B	Denosumab	Subcutaneous injection	6 month	7	Astra (14), Straumann (1)	
3	82	Female			Minodronate	Oral	4 weeks	5	Nobel Biocare (2)	
4	80	Female	4	B	Minodronate	Oral	4 weeks	11	3i (8), Straumann (1)	
5	79	Female	4	B	Risedronate	Oral	1 week	1	Nobel Biocare (5), Zimmer (3)	
									GC (2), 3i (1), Straumann (1)	
6	72	Female			Risedronate	Oral	Daily	2	Nobel Biocare (2)	MRONJ (near #35)
					Ibandronate	Oral	Daily	3		
7	72	Female			Ibandronate	Subcutaneous injection	1 month	7	Astra (3)	
8	72	Female			Risedronate	Oral	1 week	2	Platon Japan (1)	
9	67	Female			Risedronate	Oral	Daily	1	Nobel Biocare (4)	
10	86	Female	4	B	Unknown BPs	Oral	Daily	2	Nobel Biocare (2)	
11	83	Female	3	A	Alendronate	Oral	1 week	1	Nobel Biocare (3)	
12	80	Female	3	B	Alendronate	Oral	1 week	5	Nobel Biocare (5)	
13	79	Female	4	B	Alendronate	Oral	1 week	4	Astra (4)	PI-MRONJ (#14, #16, #36)
14	79	Female	2	A	Minodronate	Oral	4 weeks	8	Straumann (2), Nobel Biocare (1)	
15	76	Female			Alendronate	Oral	Daily	2	Nobel Biocare (2), POI (1)	
16	75	Female	2	A	Minodronate	Oral	4 weeks	2	Nobel Biocare (1)	
17	73	Female	2	A	Alendronate	Oral	1 week	1	Nobel Biocare (2), Endopore (2)	
									Astra (1), Straumann (1)	
18	72	Female	2	A	Alendronate	Oral	Daily	2	Nobel Biocare (1)	
19	61	Female	3	A	Alendronate	Oral	Daily	5	Nobel Biocare (1)	
20	87	Male	3	A	Minodronate	Oral	Daily	2	Nobel Biocare (1)	PI-MRONJ (#36)
					Denosumab	Subcutaneous injection	6 months	2	Straumann (2)	
21	85	Male	3	B	Unknown BP	Oral	Daily	4	Nobel Biocare (1), Endopore (1)	
22	73	Male	4	B	Denosumab	Subcutaneous injection	6 months	16	Astra (4)	

ARA, antiresorptive agent; BP, bisphosphonate; ONJ, osteonecrosis of the jaw; MRONJ, medication-related ONJ; PI-MRONJ, peri-implant MRONJ.

**Table 4 medicina-61-01348-t004:** Value of each clinical parameter.

	Baseline				Last visit				Variation			
	iPPD (mm)		iBoP (0–1)	MBL (mm)	iPPD (mm)		iBoP (0–1)	MBL (mm)	iPPD (mm)		iBoP (0–1)	MBL (mm)
	Mean	Maximum			Mean	Maximum			Mean	Maximum		
ARA group	3.1 ± 0.6	3.5 ± 1.0	0.1 ± 0.2	1.2 ± 1.1	3.8 ± 2.0	4.6 ± 2.3	0.3 ± 0.3	2.2 ± 2.5	0.7 ± 1.8	1.1 ± 2.0	0.2 ± 0.3	1.0 ± 2.2
(range, median)	(2.0–5.5, 3.2)	(2–8, 3)	(0–1, 0.2)	(0–4.5, 1.1)	(2.0–11.0, 3.2)	(2–12, 4)	(0–1, 0.2)	(0–11.0, 1.5)	(−1.0–8.0, 0.2)	(−2–8, 1)	(−1–0.8, 0)	(−1.6–10.8, 0.2)
Control group	2.8 ± 0.5	3.4 ± 0.9	0.2 ± 0.2	1.2 ± 0.8	3.2 ± 1.6	4.0 ± 1.9	0.2 ± 0.3	1.9 ± 2.1	0.3 ± 1.4	0.6 ± 1.8	0 ± 0.3	0.8 ± 1.8
(range, median)	(1.8–4.3, 2.8)	(2–8, 3)	(0–1, 0.2)	(0–2.9, 1.2)	(1.0–13.0, 2.7)	(2–13, 3)	(0–1, 0)	(0–13, 1.5)	(−1.5–8.7, 0)	(−5–7, 0)	(−0.8–0.8, 0)	(−1.3–11.5, 0.2)
Significant difference	**	ns	ns	ns	**	*	**	ns	ns	*	**	ns
*p* value	0.00155	0.23500	0.07650	0.58400	0.00700	0.01880	0.00664	0.98100	0.17400	0.03430	0.00016	0.57000
Test	Student’s t	MWUt	MWUt	Student’s t	MWUt	MWUt	MWUt	MWUt	MWUt	MWUt	MWUt	MWUt

Mean ± S.D.; ARA, antiresorptive agent; iPPD, implant probing depth; iBoP, implant bleeding on probing; MBL, marginal bone loss; MWUt, Mann-Whitney U test; ns, not significant; * *p* < 0.05, ** *p* < 0.01.

**Table 5 medicina-61-01348-t005:** Risk indicators for peri-implantitis according to logistic regression analysis.

Explanatory Variable	Unadjusted			Adjusted		
	Odds Ratio	[95% CI]	*p* Value	Odds Ratio	[95% CI]	*p* Value
ARA (yes; 1, no; 0)	3.63	[1.630–8.080]	**	3.91	[1.290000–11.9000]	*
History of insufficiency fracture (yes; 1, no; 0)	0.89	[0.386–2.050]	**	0.53	[0.172000–1.6000]	ns
History of periodontitis (yes; 1, no; 0)	8.17	[1.89000–35.400]	**	3.72	[0.743000–18.6000]	ns
MCI (class3; 1, others; 0)	1.97	[0.9420–4.130]	**	1.63	[0.574000–4.6500]	ns
Additional surgery (yes; 1, no; 0)	0.57	[0.125–2.630]	**	2.88	[0.427000–19.4000]	ns
Observational duration (over 12 years; 1, others; 0)	2.58	[1.1800–5.620]	**	2.73	[0.94000–7.9300]	ns
Variation iPPD maximum (yes; 1, no; 0)	7.42	[3.0400–18.100]	**	5.90	[2.260000–15.4000]	**
Variation iBoP mean ( yes; 1, no; 0 )	2.85	[1.35000–6.030]	**	1.86	[0.754–4.5800]	ns

CI, confidence interval; ns, not significant; ARA, antiresorptive agent; MCI, mandibular cortical index; iPPD, implant probing depth; iBoP, implant bleeding on probing * *p* < 0.05, ** *p* < 0.01; Adjusted for ARA use, history of insufficiency fracture, history of periodontitis, MCI, additional surgery, observation duration, variation iPPD maximum, variation iBoP mean.

## Data Availability

The datasets used and analyzed during the current study are available from the corresponding author upon reasonable request.

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
