# Peer review of "An Evaluation of the Peri-Implant Tissue in Patients Starting Antiresorptive Agent Treatment After Implant Placement: A Nested Case–Control Study"

_medicina, 2025, doi:10.3390/medicina61081348_

Round 1

Reviewer 1 Report

Comments and Suggestions for Authors

This is an interesting manuscript, referring to a case-control study to determine peri-implant tissue in patients starting an antiresorptive agent. It has a strength: the periodontal reclassification of patients according to the current classification.

In essence, a grammatical review of the manuscript is recommended. It detects spelling errors and even redundancy (e.g., "aging of society," line 51; little attention has been “paid” to the impact of ARA use on... line 75).

The wording of lines 52-53 should be reconsidered. "The etiology, pathogenesis, and treatment of MRONJ are still unclear, and evidence is still being accumulated." It is self-contradictory, based on what you write on next lines 53-55.

Review the inclusion of the meaning of all acronyms in the abbreviations list, e.g., "DPR," line 152.

In Table 4, how the maximum variation in iPPD (mm) was obtained, it should be indicated in section 2.4.

The statistical analysis presented in Table 4 was of the problem group versus the control group; however, it is an opportunity for improvement to consider the baseline measurement versus the last visit. The Variation results column is illustrative, but a statistical analysis would be more rigorous and thus present a better argument, correlating with the results presented in Table 5.

Author Response

Reviewer 1 Comments

This is an interesting manuscript, referring to a case-control study to determine peri-implant tissue in patients starting an antiresorptive agent. It has a strength: the periodontal reclassification of patients according to the current classification.

Response: AGREE

Dear Editor, Thank you for your thoughtful comment. We appreciate your understanding of our intention to focus on periodontal disease classification as a topic.

In essence, a grammatical review of the manuscript is recommended. It detects spelling errors and even redundancy (e.g., "aging of society," line 51; little attention has been “paid” to the impact of ARA use on... line 75).

Response: AGREE

Dear Editor, Thank you for the detailed review. We have corrected the points reviewers pointed out.

Change was made:

P2L50-51, P2L75-78

The wording of lines 52-53 should be reconsidered. "The etiology, pathogenesis, and treatment of MRONJ are still unclear, and evidence is still being accumulated." It is self-contradictory, based on what you write on next lines 53-55.

Response: AGREE

Dear Editor, Thank you for your feedback. We have revised the wording.

Change was made:

The following text has been changed.

“Although the etiology, pathophysiology, and treatment of MRONJ remain unclear, these topics are gradually being elucidated.” (P3L51-53)

Review the inclusion of the meaning of all acronyms in the abbreviations list, e.g., "DPR," line 152.

Response: AGREE

Dear Editor, Thank you for your detailed review. We have added explanations for two abbreviations.

Change was made:

Added PICO and DPR to the abbreviations section (P2L83-84, P12L367).

In Table 4, how the maximum variation in iPPD (mm) was obtained, it should be indicated in section 2.4.

Response: AGREE

Dear Editor, Thank you for your constructive comments. We changed it to a sentence that was easier to understand, as the reviewer advised.

Change was made:

The following text has been changed.

“The mean and maximum iPPD, iBoP, and MBL were measured at baseline and at the final visit. For all parameters, changes in the mean and maximum values over time were cal-culated by subtracting the baseline values from the values at the final visit..” (P4L151-154)

The statistical analysis presented in Table 4 was of the problem group versus the control group; however, it is an opportunity for improvement to consider the baseline measurement versus the last visit. The Variation results column is illustrative, but a statistical analysis would be more rigorous and thus present a better argument, correlating with the results presented in Table 5.

Response: AGREE

Dear Editor, Thank you for your constructive comments. We agree with the opinion. We have added new results in the text.

Change was made:

P9L262-266

“The changes from baseline to the last visit were 2.9 ± 0.5 mm to 3.4 ± 1.8 mm for the mean iPPD, 3.5 ± 1.0 mm to 4.3 ± 2.1 mm for the maximum iPPD, 0.2 ± 0.2 to 0.2 ± 0.3 for the mean iBoP, and 1.2 ± 0.9 mm to 2.1 ± 2.3 mm for the MBL. The Mann-Whitney U-test showed that there were significant increases from baseline to the last visit in the maxi-mum iPPD (P = 0.0006) and MBL (P = 0.0003).”

Reviewer 2 Report

Comments and Suggestions for Authors

The authors Sike et al. have conducted detailed research on the effect of antiresorptive agents on medication-related osteonecrosis of the jaw (MRONJ)since 2021. Their findings are reported in published articles and are referred to in the literature. 

The study is based on 61 patients (192 implants), a relatively small sample size, and was conducted over 2012–2024.

Focus: Patients starting antiresorptive therapy (ARA) after implant placement.

Conclusion: Importance of monitoring peri-implantitis and implant pocket depth due to changing health/medication status.

Study Scope is Limited:

Conducted in a specific, possibly single-center setting.

Lacks diverse population sampling (only 61 patients; majority female).

No data provided on ethnicity, socioeconomic status, or regional healthcare practices, which are essential for demographic extrapolation.

MRONJ Development is Multifactorial:

Influenced by type and duration of antiresorptive therapy (e.g., bisphosphonates, denosumab).

Risk factors: age, comorbidities (e.g., diabetes, cancer), dental hygiene, and smoking.

These vary significantly across populations and healthcare systems (e.g., US vs. EU vs. Japan).

Regional Differences in ARA Use:

Prescription patterns, dental care access, and implant protocols differ across countries.

        •  

Epidemiological Data Required:

Predicting regional impact requires large-scale, multicenter cohort studies.

Databases like those from CDC (USA), EMA (EU), or NIH would be better suited for such extrapolation.

The authors' findings are valuable 

  • They highlight a clinical risk that may apply universally: starting ARA treatment after implant placement requires closer monitoring for MRONJ signs.

  • Their emphasis on peri-implantitis and pocket depth as early warning signs is clinically translatable even outside the study region.

Author Response

Reviewer 2 Comments

Conducted in a specific, possibly single-center setting.

Response: AGREE

Dear Editor, Thank you for your detailed review. As Reviewer 2 pointed out, this study was conducted in a single-center setting. Please check the section on Study design and participants (P3L97-99), which states Nihon University School of Dentistry Dental Hospital. Thank you for reading carefully.